# Catalytic Degradation of Toluene over MnO₂/LaMnO₃: Effect of Phase Type of MnO₂ on Activity

**Lu Li** [1,*], **Yuwei Liu** [2], **Jingyin Liu** [2], **Bing Zhou** [2], **Mingming Guo** [3] and **Lizhong Liu** [2,*]

1   State Key Laboratory of Electrical Insulation and Power Equipment, Center of Nanomaterials for Renewable Energy, School of Electrical Engineering, Xi'an Jiaotong University, Xi'an 710049, China
2   School of Chemistry and Chemical Engineering, Nantong University, 9, Seyuan Road, Nantong 226019, China
3   School of Environmental Science and Engineering, Shanghai Jiao Tong University, 800, Dong Chuan Road, Shanghai 200240, China
*   Correspondence: dorislee1224@xjtu.edu.cn (L.L.); lzliu@ntu.edu.cn (L.L.)

**Abstract:** Series of $\alpha$, $\beta$, $\gamma$, $\delta$ type MnO₂ supported on LaMnO₃ perovskite was developed by a one-pot synthesis route. Compared with $\alpha$-MnO₂, $\beta$-MnO₂, $\gamma$-MnO₂, $\delta$-MnO₂ and LaMnO₃ oxides, all MnO₂/LaMnO₃ showed promotional catalytic performance for toluene degradation. Among them, $\alpha$-MnO₂/LaMnO₃ holds the best active and mineralization efficiency. By the analysis of N₂ adsorption-desorption, XPS and H₂-TPR, it can be inferred that the improved activity should be ascribed to the higher proportion of lattice oxygen, better low-temperature reducibility and larger specific surface area. Besides, the byproducts from the low-temperature reaction of toluene oxidation were detected by a TD/GC-MS, confirming the presence of the intermediates. Combined with the in-situ DRIFTS, the catalytic degradation path of toluene oxidation has also been discussed in depth.

**Keywords:** phase type; perovskite; toluene; catalytic oxidation; degradation path





## 1. Introduction

Volatile organic compounds (VOCs) are harmful pollutants released into the environment resulting from a variety of commercial, industrial, and domestic practices [1–3]. Among them, toluene is a ubiquitous aromatic VOC and an important contributor to the formation of ozone (O₃) as well as a secondary organic aerosol (SOA) [2,4]. Meanwhile, toluene is easily absorbed by the skin and mucous membranes, which has devastating effects on human organs and metabolic systems [5]. Many traditional technologies, such as condensation, membrane separation, absorption, thermal incineration, and catalytic oxidation, have been utilized to control VOCs [6–8], wherein catalytic oxidation has been believed to be one of the most effective strategies, which could completely convert pollutants into harmfulness final products, carbon dioxide and water [9]. Catalysts are the key factor in catalytic oxidation technology. Generally, noble metal and non-precious metal catalysts are the two major types of materials applied in toluene catalytic combustion. However, noble metals, such as gold, palladium, platinum, ruthenium, etc., are easily restricted by their high cost, sintering, and poisoning in industrial applications [10]. On the contrary, non-noble mixed metal oxides have been considered universal candidates due to their wide source, low price, relatively high activity, and good anti-toxicity [11,12].

Metal oxides/perovskites (MOₓ/ABO₃) are a typical class of non-noble mixed metal oxide compounds, especially Mn-based MnO₂/AMnO₃, which have been widely used in the fields of denitration, desulfurization and VOCs oxidation [13–15]. Si et al. synthesized a $\gamma$-MnO₂/LaMnO₃ catalyst by selectively removing surficial La cation from LaMnO₃ and obtained a superior catalytic performance on toluene oxidation [16]. Yang et al. studied the LaMnO₃ perovskite assembled by $\delta$-MnO₂ through a gunpowder-like combustion method and pointed out that the boosted active phase-carrier interplays, improved redox

abilities led to good performance for toluene oxidation [17]. We also synthesized a $\gamma$-$MnO_2$/$SmMnO_3$ material by in-situ acid etching of $SmMnO_3$ and found that the composite of $\gamma$-$MnO_2$ was successfully exposed on the surface of $SmMnO_3$, which improved the physicochemical properties, thus promoted toluene oxidation capacity [18]. Kim et al. [19] paid attention to different manganese valent oxides $Mn_3O_4$, $Mn_2O_3$, and $MnO_2$ in the catalytic combustion of benzene and toluene. Figueredo et al. [20] also made a comparison of different manganese oxides $Mn_3O_4$, $Mn_2O_3$, and their biphasic compound $Mn_3O_4$/$Mn_2O_3$ material on the catalytic evaluation of ethene and propene. However, little attention is paid to digging out the effect of different phase types of manganese dioxide supported on perovskites, as well as the degree of activating lattice oxygen and catalytic activity. Meanwhile, a simple modulation process of $MnO_2$ on the perovskite surface is also particularly important due to the practical point of view. Besides, the catalytic mechanism of catalysts and the degradation path of toluene still require great efforts to clarify.

Based on the above challenges, herein we provide a relatively simple synthesis route for $MnO_2$/$LaMnO_3$ preparation, in which the phase type ($\alpha$, $\beta$, $\delta$, and $\gamma$) of $MnO_2$ can be directly adjusted by controlling the proportion of manganese source precursors. Meanwhile, the activation degree of lattice oxygen on the surface could also be controlled by adjusting the surficial phase type of $MnO_2$ of $LaMnO_3$. Combined with the characterizations of XRD, SEM, $N_2$ physisorption, XPS, $H_2$-TPR, etc., the influence of the phases of $MnO_2$ in $LaMnO_3$ on the physical and chemical properties and the catalytic activities for toluene degradation were explored, as well as the internal relationship of structure and catalytic behavior was also in-depth discussion. Finally, the types of byproducts by the low-temperature catalytic oxidation of toluene and the degradation path of toluene oxidation were discussed.

## 2. Results and Discussion

### 2.1. Morphology and Crystal Phase Structure

The crystal structures of samples were characterized by XRD. As shown in Figure S1, the diffraction peaks at $2\theta$ = 22.94°, 32.78°, 38.47°, 40.16°, 40.64°, 46.84°, 52.62°, 52.96°, 58.10°, 58.70°, 68.02, 68.70 and 78.10°, corresponding to the (012), (110), (113), (202), (006), (024), (122), (116), (214), (018), (220), (208), and (128) planes of $LaMnO_3$ (PDF# 50-0298) [21]. Meanwhile, as illustrated in Figure 1, it was found that the new peaks of $\alpha$-MO/LMO (Figure 1a) at $2\theta$ = 28.84° (310) and 37.5° (211), $\beta$-MO/LMO (Figure 1b) at $2\theta$ = 28.68° (110), 37.32° (101), and 42.8° (111), $\gamma$-MO/LMO (Figure 1c) at $2\theta$ = 22.4° (120), 34.5° (031), 37.14° (131), and 57.4° (160), and $\delta$-MO/LMO (Figure 1d) at $2\theta$ = 12.68° (001), and 25.2° (002), corresponded to the standard cards of PDF# 44-0141 ($\alpha$-$MnO_2$) [22], PDF# 24-0735 ($\beta$-$MnO_2$) [23], PDF# 80-1098 ($\delta$-$MnO_2$) [24], and PDF# 14-0644 ($\gamma$-$MnO_2$) [18], respectively, showing that the different phases types ($\alpha$, $\beta$, $\delta$, and $\gamma$) $MnO_2$ were successfully combined with $LaMnO_3$ perovskite. In addition, the full width at half maximum (FWHM) of the characteristic peaks at $2\theta$ = 32.78° are 0.38° for LMO, 0.37° for $\alpha$-MO/LMO, 0.31° for $\beta$-MO/LMO, 0.29° for $\delta$-MO/LMO and 0.28° for $\gamma$-MO/LMO. Obviously, the FWHM at $2\theta$ = 32.78° of $\alpha$-MO/LMO is closer to that of LMO, indicating that the composite degree of $\alpha$-$MnO_2$ on the surface of $LaMnO_3$ is better, which is beneficial to inhibit the further agglomeration of $LaMnO_3$ during the hydrothermal reaction.

Figure 2a–j shows the SEM images of LMO, $\alpha$-MO/LMO, $\beta$-MO/LMO, $\delta$-MO/LMO and $\gamma$-MO/LMO. As can be seen in Figure 2a,b, the morphology of the LMO sample exhibits a bulk structure with large pore sizes, indicating that the high-temperature preparation process caused the agglomeration of the LMO catalyst. When the $\alpha$-$MnO_2$ was combined with $LaMnO_3$, the composite showed an obvious porous structure (Figure 2c), and the surface of the $LaMnO_3$ framework was covered with a furry substance (Figure 2d), which should be the $\alpha$-$MnO_2$ with stacking-nanoneedle structure [23]. These changes may imply an increase in the specific surface area of the catalyst, which is conducive to the contact between toluene molecules and the catalyst, promoting the catalytic oxidation of toluene. In addition, it can be observed from the SEM images of $\beta$-MO/LMO in Figure 2e,f that the morphology of $\beta$-MO/LMO is disorganized and a small amount of sheet-like structures

are mixed in the sample, indicating that the presence of β-MnO$_2$ in the composite [23,25]. Similarly, the large-scale flower-like structures are also found in Figure 2g,h, confirming the formation of δ-MnO$_2$ in the δ-MO/LMO sample [26,27]. However, when γ-MnO$_2$ and LaMnO$_3$ were combined, lots of ca. 50 nm nano-spherical particles and large-sized agglomerates appeared in the SEM image (Figure 2i,j). Combined with the analysis of XRD, it can be inferred that the nano-spherical particles should belong to γ-MnO$_2$.

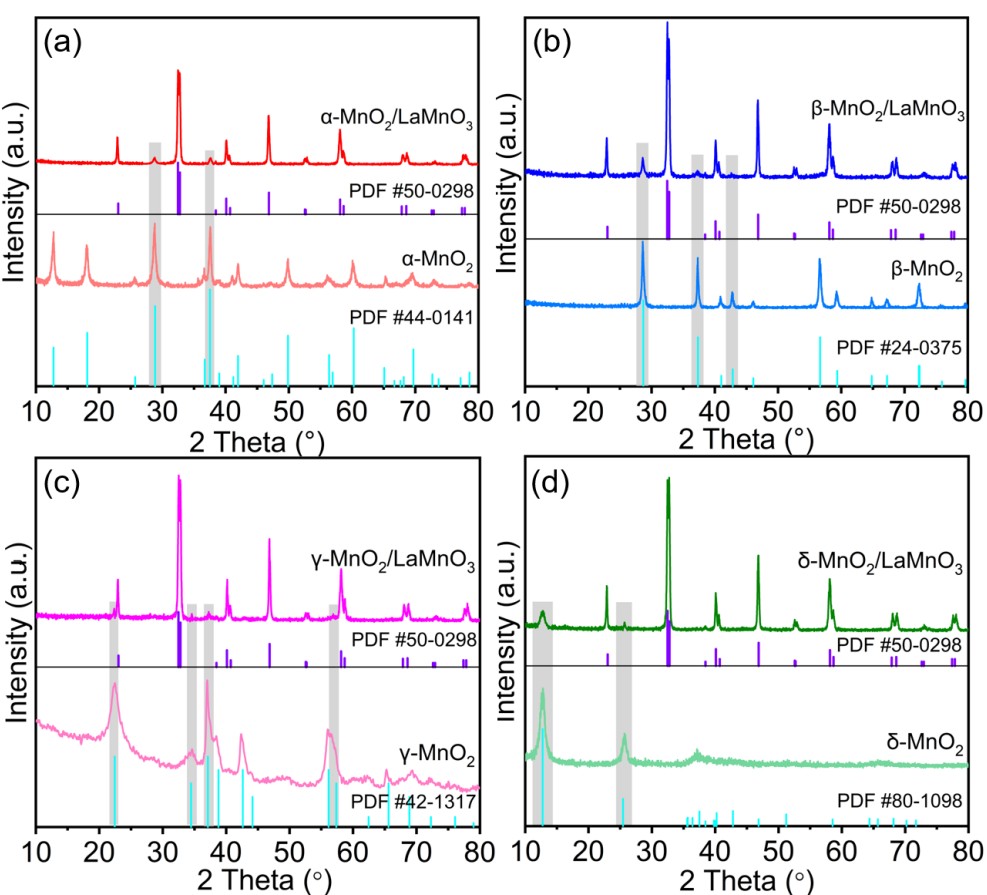

**Figure 1.** XRD patterns of (**a**) α-MO/LMO, α-MnO$_2$ and corresponding standard PDF card diffraction profile; (**b**) β-MO/LMO, β-MnO$_2$ and corresponding standard PDF card diffraction profile; (**c**) γ-MO/LMO γ-MnO$_2$ and corresponding standard PDF card diffraction profile; and (**d**) δ-MO/LMO, δ-MnO$_2$ and corresponding standard PDF card diffraction profile.

### 2.2. Catalytic Performance

Figure 3a demonstrates the toluene oxidation over LMO, α-MO/LMO, β-MO/LMO, δ-MO/LMO, and γ-MO/LMO. Before testing the properties of all samples, the blank experiment was performed without a catalyst, and the conversion was not found (<320 °C), indicating that there was no occurrence of homogeneous reactions under the adopted reaction [18,28]. The catalytic activities of all samples hoisted with the reaction temperature and the heightened sequence were as follows: LMO < γ-MO/LMO < δ-MO/LMO < β-MO/LMO < α-MO/LMO. Obviously, after loading manganese dioxide, the performance of new catalysts improved. Moreover, it can be seen from Figure S2 that the activity of each composite was also higher than that of the corresponding pure MnO$_2$. In order to further analyze the catalytic activities of all catalysts, the $T_{90}$, and $T_{50}$ values (reaction temperature vs. conversion of 50%, and 90%) were used to further assess their performance. As listed in Table 1, the T$_{90}$, and T$_{50}$ values were 260, and 237 °C for α-MO/LMO, 289, and 246 °C for β-MO/LMO, 294, and 255 °C for δ-MO/LMO, 316, and 261 °C for γ-MO/LMO, >320, and 300 °C for LMO, 295, and 252 °C for α-MO, > 320, and 302 °C for β-MnO$_2$, > 320, and

289 °C for δ-MO, and > 320, and 286 °C for γ-MO, respectively. Obviously, α-MO/LMO holds lower $T_{90}$, and $T_{50}$ values, indicating a better performance for toluene oxidation.

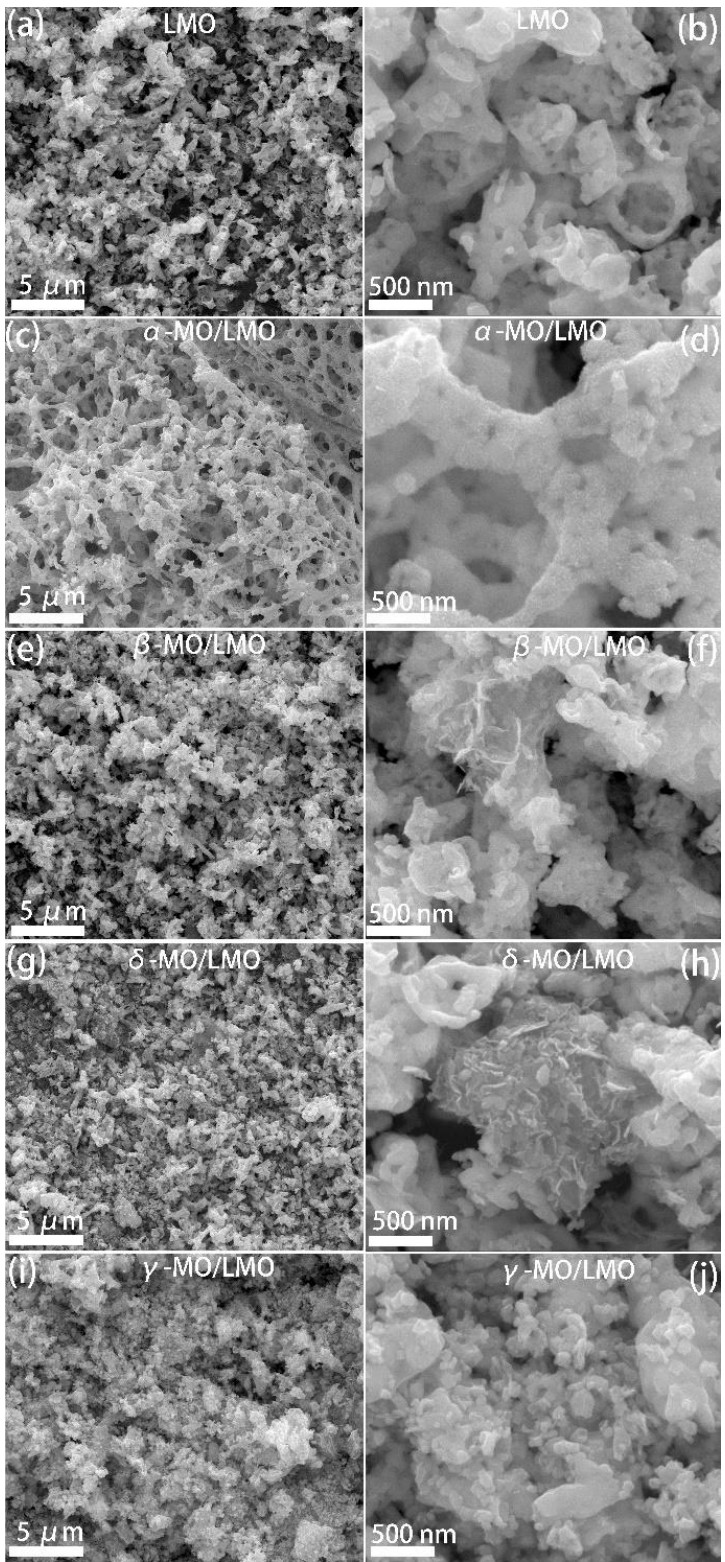

**Figure 2.** SEM images of (**a**,**b**) LMO, (**c**,**d**) α-MO/LMO, (**e**,**f**) β-MO/LMO, (**g**,**h**) δ-MO/LMO, and (**i**,**j**) γ-MO/LMO.

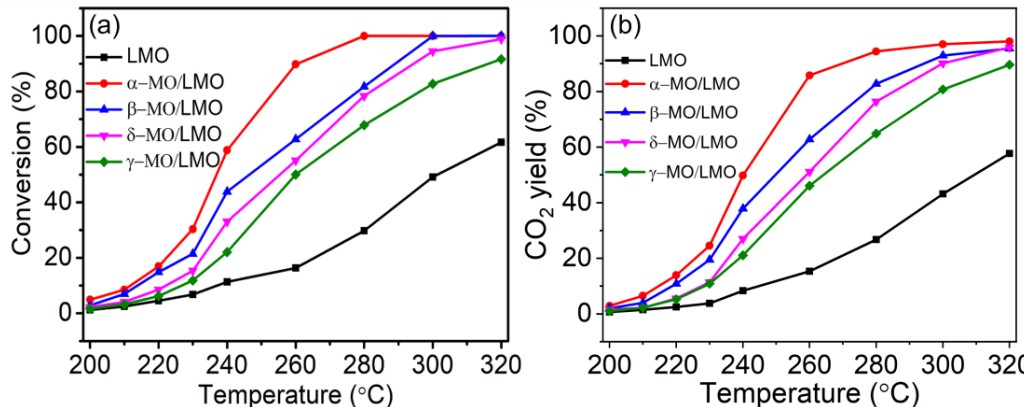

**Figure 3.** (**a**) Toluene conversion vs. temperature and (**b**) $CO_2$ yield over LMO, α-MO/LMO, β-MO/LMO, δ-MO/LMO, and γ-MO/LMO.

**Table 1.** Catalytic performance for toluene oxidation over various catalysts.

| Samples | Conversion | | CO₂ Yield | | *r* |
|---|---|---|---|---|---|
| | $T_{90}$ (°C) | $T_{50}$ (°C) | $T_{90}$ (°C) | $T_{50}$ (°C) | ($10^{-10}$ mol m$^{-2}$ s$^{-1}$) |
| α-MO/LMO | 260 | 237 | 269 | 241 | 3.13 |
| β-MO/LMO | 289 | 246 | 293 | 250 | 2.70 |
| δ-MO/LMO | 294 | 255 | 299 | 259 | 1.85 |
| γ-MO/LMO | 316 | 261 | 320 | 264 | 1.73 |
| LMO | >320 | 300 | >320 | 309 | 2.62 |
| α-MO | 295 | 252 | / | / | / |
| β-MO | >320 | 302 | / | / | / |
| δ-MO | >320 | 289 | / | / | / |
| γ-MO | >320 | 286 | / | / | / |

Since the specific surface areas ($S_{BET}$) are quite different (shown in Table 2) and thermal effect may also exist at higher DCE conversion, the apparent reaction rates of these catalysts are calculated at low temperatures (210 °C) to reflect their inherent catalytic activities. As listed in Table 1, the values decrease in the sequence of α-MO/LMO > β-MO/LMO > LMO > δ-MO/LMO > γ-MO/LMO, which is basically consistent with the order for conversion efficiency, while LMO showed a higher apparent reaction rate than that of δ- and γ- phased samples.

**Table 2.** Surface element compositions, surface area and reducibility of LMO, α-MO/LMO, β-MO/LMO, δ-MO/LMO and γ-MO/LMO.

| Catalysts | $S_{BET}$/ m²·g⁻¹ | Molar Ratios of Surface Elements by XPS | | H₂ uptake/mmol·g⁻¹ (50–450 °C) |
|---|---|---|---|---|
| | | $Mn^{4+}/Mn^{3+}$ | $O_{latt}/O_{ads}$ | |
| LMO | 14.0 | 0.38 | 1.21 | 2.03 |
| α-MO/LMO | 42.3 | 0.52 | 2.48 | 4.09 |
| β-MO/LMO | 40.0 | 0.46 | 2.39 | 3.51 |
| δ-MO/LMO | 39.6 | 0.42 | 2.12 | 3.03 |
| γ-MO/LMO | 35.6 | 0.40 | 1.89 | 2.44 |

To better evaluate the destruction ability of prepared catalysts, the mineralization efficiency of toluene oxidation was also measured. As shown in Figure 3b and Table 1, the mineralization temperatures of these catalysts for toluene are higher than that of conversion, where the variation trend of $CO_2$ yield was similar to that of conversion. Besides, α-MO/LMO sample still maintains the best behavior in $CO_2$ generation capacity.

### 2.3. Textual Structure and Surface Area

The $N_2$ adsorption-desorption isotherms of LMO, α-MO/LMO, β-MO/LMO, δ-MO/LMO and γ-MO/LMO are shown in Figure 4a, and the specific surface areas are listed in Table 2. As displayed in Figure 4a, the isotherms of LMO, α-MO/LMO, β-MO/LMO, δ-MO/LMO and γ-MO/LMO presented a typical IUPAC type II pattern with non-uniform slit pores (H3 type) of the hysteresis loop, signifying the existence of macroporous and mesoporous structures [11,18,29]. In addition, it can also be observed from the data listed in Table 2 that the specific surface areas of LMO, α-MO/LMO, β-MO/LMO, δ-MO/LMO, and γ-MO/LMO were 14.0, 42.3, 40.0, 39.6, and 35.6 $m^2 \cdot g^{-1}$, respectively. Compared with pure perovskite, the specific surface areas of various phase types of $MnO_2/LaMnO_3$ were all increased. The high specific surface area is beneficial to promote the oxidation of toluene molecules on the surface of the catalyst [29,30]. Namely, the α-MO/LMO catalyst with the highest specific surface area may result in better activity [31].

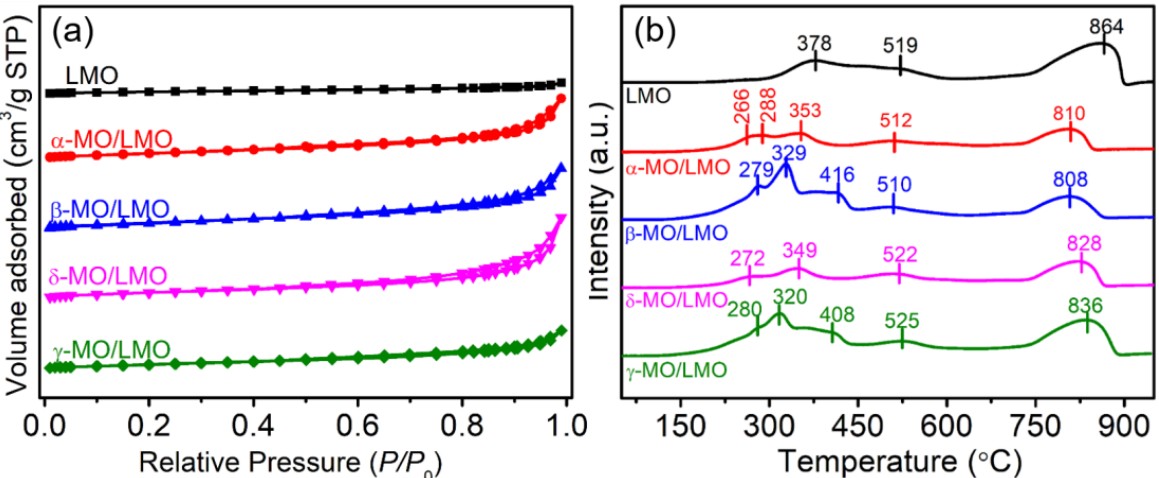

**Figure 4.** (**a**) $N_2$ adsorption-desorption isotherms and (**b**) $H_2$-TPR profiles of LMO, α-MO/LMO, β-MO/LMO, δ-MO/LMO and γ-MO/LMO.

### 2.4. Mn Oxidized State and Oxygen Species

The Mn-oxidized state and oxygen species on the surface of LMO, α-MO/LMO, β-MO/LMO, δ-MO/LMO, and γ-MO/LMO were obtained using XPS. The Mn $2p_{3/2}$, O $1s$ XPS spectra of all catalysts are shown in Figure 5a,b and the quantitative analysis is presented in Table 2. As displayed in Figure 5a, the components divided from the Mn $2p_{3/2}$ XPS spectra at 643.58, 641.78, and 640.46 eV corresponded to $Mn^{4+}$, $Mn^{3+}$, and $Mn^{2+}$, respectively [11,32–34]. Compared with that of pure LMO perovskite, the Mn $2p$ binding energy of MO/LMO is relatively reduced, suggesting that the density of the extranuclear electron cloud of the Mn element is increased. In fact, the asymmetric charge density distribution of Mn showed a significant increase in the hybridization of Mn $3d$ and O $2p$ orbitals and shortening of Mn-O bond length [21], which may be helpful to activate the lattice oxygen and improve the reducibility of the catalysts [12]. The increase in the molar ratio of $Mn^{4+}/Mn^{3+}$ in the order of LMO (0.38) < γ-MO/LMO (0.40) < δ-MO/LMO (0.42) < β-MO/LMO (0.46) < α-MO/LMO (0.52), which is consistent with the order of their activities. The increase in the proportion of high-valent manganese implies the lattice oxygen on the surface is more easily activated. The activated lattice oxygen can migrate to the oxygen vacancies and then transform into overflowed adsorbed oxygen species easily to oxidize toluene [21]. Consequently, the α-MO/LMO with the high molar ratio of $Mn^{4+}/Mn^{3+}$ has a higher toluene oxidation potential.

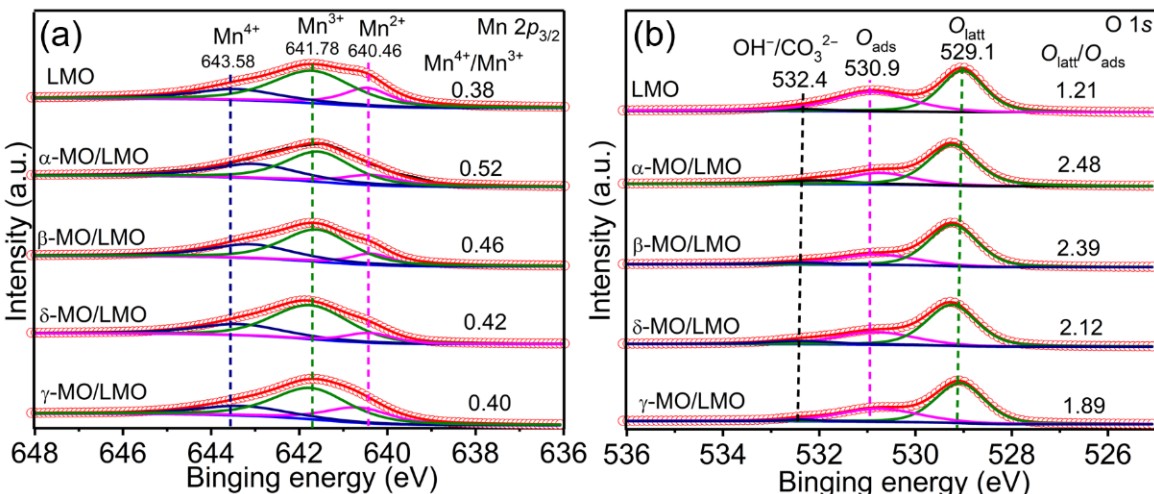

**Figure 5.** (**a**) Mn $2p_{3/2}$, and (**b**) O 1$s$ XPS spectra of LMO, α-MO/LMO, β-MO/LMO, δ-MO/LMO and γ-MO/LMO.

Furthermore, the oxygen species on the surface of all catalysts were determined by XPS. As shown in Figure 5b, the peaks of O 1$s$ XPS spectra located at 529.1, 530.9, and 532.4 eV are attributed to the lattice oxygen ($O_{latt}$), surface adsorbed oxygen ($O_{ads}$) and surface carbonate ($CO_3^{2-}$) or hydroxide ($OH^-$), respectively [18,23,28]. In contrast to that of Mn 2$p$, the O 1$s$ binding energy of MO/LMO is relatively elevated, explaining that the density of the extranuclear electron cloud of O element cut down, facilitating the activation of lattice oxygen. As listed in Table 2, the order of the $O_{latt}/O_{ads}$ molar ratio enhanced in the arrangement of LMO (1.21) < γ-MO/LMO (1.89) < δ-MO/LMO (2.12) < β-MO/LMO (2.39) < α-MO/LMO (2.48), which is consistent with the increasing sequences of $Mn^{4+}/Mn^{3+}$ ratio and toluene conversion. The results testified that the composite of $MnO_2$ and $LaMnO_3$ promoted the enhancement of the activated lattice oxygen, and the surface of α-MO/LMO possessed the most $O_{latt}$, which induced the best catalytic oxidation performance of toluene.

*2.5. Reducibility*

The reducibility of LMO, α-MO/LMO, β-MO/LMO, δ-MO/LMO, and γ-MO/LMO was evaluated by $H_2$-TPR. As shown in Figure 4b, the peaks for LMO at 378 °C are attributed to the restituting of $Mn^{4+}$, and the peak at 512 °C was assigned to the single-electron reduction of $Mn^{3+}$ located in a coordination-unsaturated microenvironment, and the peak at 864 °C is classified to the retrieval of the left $Mn^{3+}$ [18,24,26]. For α-MO/LMO, the peaks at 266 and 288 °C should correspond to the reduction of surface $Mn^{4+}$ and deep $Mn^{4+}$; the peaks at 353 and 512 °C belong to the reduction of surface $Mn^{3+}$ and deep $Mn^{3+}$; the peaks at 810 °C is pertain to the lowering of the left $Mn^{3+}$ in LMO. Analogously, the peaks of β-MO/LMO at 279 and 329 °C belong to the surface $Mn^{4+}$ and deep $Mn^{4+}$, 416 and 510 °C corresponding to the reduction of surface $Mn^{3+}$ and deep $Mn^{3+}$, 808 °C corresponding to the left $Mn^{3+}$ in LMO; the peaks of δ-MO/LMO at 272 °C corresponding to the reduction of $Mn^{4+}$, 349 and 522 °C corresponding to the reduction of surface $Mn^{3+}$ and deep $Mn^{3+}$, 828 °C corresponding to the left $Mn^{3+}$ in LMO; the peaks of γ-MO/LMO at 280 and 320 °C belong to the surface $Mn^{4+}$ and deep $Mn^{4+}$, 408 and 525 °C corresponding to the reduction of surface $Mn^{3+}$ and deep $Mn^{3+}$, 836 °C corresponding to the left $Mn^{3+}$ in LMO. Obviously, α-MO/LMO has better reducibility at low temperatures. The $H_2$ consumption of each catalyst was calculated, and it can be confirmed from Table 2 that the order of hydrogen consumption between 50 and 450 °C is as follows: LMO (2.03 mmol·g$^{-1}$) < γ-MO/LMO (2.44 mmol·g$^{-1}$) < δ-MO/LMO (3.03 mmol·g$^{-1}$) < β-MO/LMO (3.51 mmol·g$^{-1}$) < α-MO/LMO (4.09 mmol·g$^{-1}$). Namely, the α-MO/LMO hold the high $H_2$-uptake, and the more $H_2$-uptake represents a higher valence of Mn [22,26], which indirectly shows that α-MO/LMO has more activated lattice oxygen on the surface, leading to the better toluene

oxidation. Combined with the results of H$_2$-TPR, XPS, BET and their activities, α-MO/LMO has a higher proportion of lattice oxygen, better low-temperature reducibility and larger surface area in comparison with other catalysts, which leads to the best catalytic activity.

### 2.6. Possible Degradation Mechanism

The rare content of exhaust byproducts was identified by the thermal desorption/gas chromatograph mass spectrometer instrument. As shown in Figure 6, trace byproducts from the $T_{50}$ condition of α-MO/LMO oxidizing toluene illustrated some identifiable organic byproducts, such as benzene, benzoic acid, benzaldehyde, 1,3-diethenyl-benzene, acetic acid, benzyl alcohol, phenol, 4-phenyl-3-buten-2-one, octanoic acid, cyclohexasiloxane, tridecane, nonanoic acid, n-decanoic acid, benzophenone, and benzyl benzoate. In fact, toluene oxidation involves the surface active oxygen species excited by the heating [5,31,35]. Furthermore, the intermediates for the catalytic oxidation of toluene over α-MO/LMO were obtained and displayed in Figure 7. A series of structural changes in the surface of α-MO/LMO occurred. With the temperature change, the peak at 3070 cm$^{-1}$ belongs to the C-H of vinyl stretching vibration [36]. The peaks at 1596, 1533 and 1419 cm$^{-1}$ pertained to the C-C and C=O stretching vibration and carbonate species [37,38]. Moreover, the vibration modes at 1288 cm$^{-1}$ and 1038 cm$^{-1}$ belong to $v_s$(COO) stretching vibration [39]. The results showed that toluene was oxidized to some carbonyl-containing species. Even though a higher degradation efficiency was found for α-MO/LMO catalysts and only trace organic byproducts were detectable in the outgas, some of the products would cause more toxicity to human health or the environment [40], even in a very small quantity. The comprehensive assessment of these trace byproducts would be our future research direction to help the choice of new-dedicated catalysts for the VOC oxidation technique.

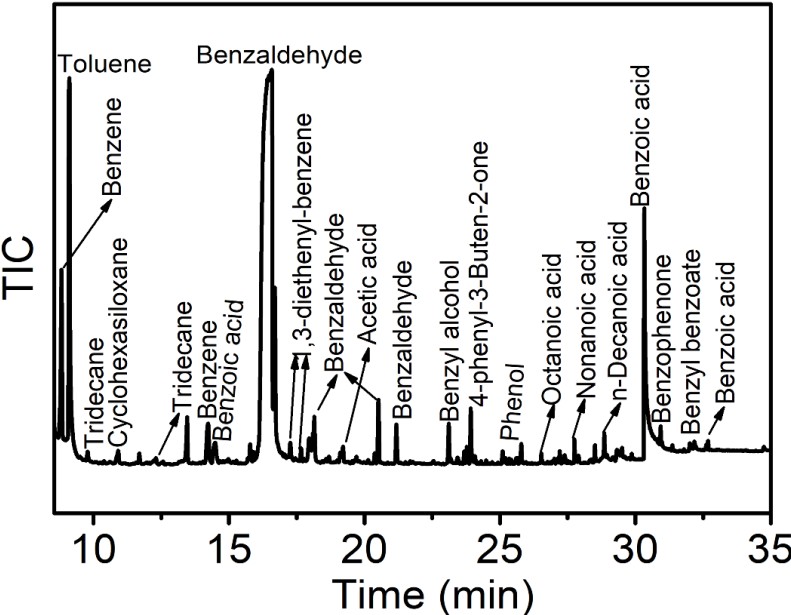

**Figure 6.** GC-MS spectrum of toluene oxidation over α-MO/LMO.

Based on the characterization of GC-MS and in-situ DRIFTS, the toluene oxidation over α-MO/LMO should adhere to the Mars-van Krevelen mechanism [41,42]. The active oxygen species on the surface of α-MO/LMO include adsorbed oxygen ($O_{ads}$) and lattice oxygen ($O_{latt}$). At a temperature, the activated $O_{ads}$ and $O_{latt}$ can be transformed into each other. Meantime, the consumed active oxygen can be repaired by the interaction of O$_2$ molecules in the atmosphere with oxygen vacancies. On account of a relatively low temperature, the number of excited reactive oxygen species on the surface of α-MO/LMO was limited, resulting in insufficient toluene oxidation, which can bring about a series of organic byproducts [18,21,43]. The toluene was decomposed and may undergo a series

of consequent reactions [18,24]: toluene → benzyl alcohol → benzaldehyde → benzoic acid → benzene → acetic acid → $CO_2$. Wherefore, it is necessary for toluene oxidation to further heighten the reaction temperature to manufacture more active oxygen species.

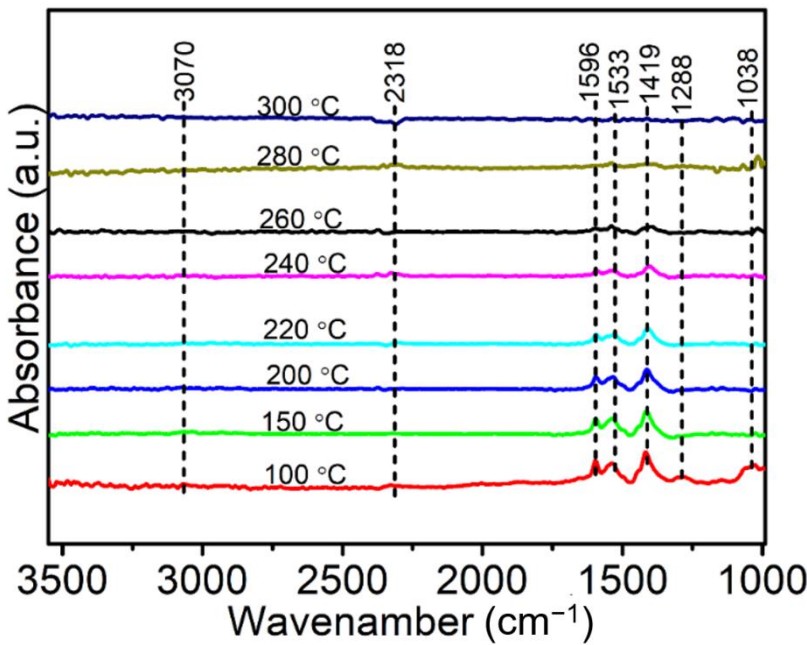

**Figure 7.** In-situ DRIFTS result of toluene oxidation over α-MO/LMO.

## 3. Experimental Section

### 3.1. Synthesis of Catalysts

LaMnO$_3$ perovskite was first prepared via direct calcination without adding water as solvent of metal salts and citric acid, which was similar to the preparation we previously reported [44]. α-MnO$_2$/LaMnO$_3$, β-MnO$_2$/LaMnO$_3$, δ-MnO$_2$/LaMnO$_3$, γ-MnO$_2$/LaMnO$_3$ (recorded as α-MO/LMO, β-MO/LMO, δ-MO/LMO, γ-MO/LMO) were synthesized by a simple one-step method, with the same molar ratio of MnO$_2$ to LaMnO$_3$ is 3:7. The typical synthesis procedure of α-MO/LMO was as follows: 0.35 g of KMnO$_4$ and 0.15 g of MnSO$_4$·H$_2$O was dissolved in 60 mL deionized water. After forming an emulsion-like, 2 g of LaMnO$_3$ powders were added to the above-mixed liquid and stirred for 30 min, and then transferred to a 100 mL PTFE kettle for hydrothermal reaction at 160 °C for 12 h. Ultimately, the acquired precipitates were calcined at 400 °C for 4 h.

For β-MO/LMO, 0.528 g of MnSO$_4$·H$_2$O and 0.712 g of (NH$_4$)$_2$S$_2$O$_8$ were added to 60 mL of deionized water. Other steps are similar to that of α-MO/LMO. For δ-MO/LMO, the addition amounts of KMnO$_4$ and MnSO$_4$·H$_2$O were 0.42 g and 0.78 g, respectively, and other steps were similar to that of α-MO/LMO. For γ-MO/LMO, 2 g of LaMnO$_3$, 0.3169 g of MnSO$_4$·H$_2$O, and 1 mL of concentrated H$_2$SO$_4$ were added to 60 mL deionized water and agitated at 80 °C for 30 min, then added 0.948 g of KMnO$_4$ to keep stirring for 24 h. The acquired precipitate was calcined at 400 °C for 4 h.

### 3.2. Characterization

The prepared samples were characterized by various measurement techniques, X-ray diffraction (XRD, Shimadzu, Tokyo, Japan), Scanning electron microscope (SEM, Bruker, Berlin, Germany), X-ray photoelectron spectra (XPS, Perkin-Elmer, Waltham, MA, USA), and H$_2$ temperature-programmed reduction (H$_2$-TPR, Micromeritics, Norcross, GA, USA). All the character processes and instrument parameters are detailed in the Supplementary Material document.

### 3.3. Catalytic Evaluation

The catalytic oxidation of toluene over all samples was estimated in a successive flow fixed bed quartz micro-reactor (I.D. = 0.006 m). 0.20 g of samples (80–120 mesh) were applied to investigate their properties. The air stream with approximately 1000 ppm of toluene crossed over the bed with a total flow rate of 40 mL/min, with a weight hourly space velocity (WHSV) of 12,000 mL/(g h$^{-1}$). The inset and outset concentration of toluene was detected by the GC with a flame ionization detector (FID).

The conversion of toluene is calculated by the following equations:

$$X = \frac{C_{in} - C_{out}}{C_{in}} \times 100\% \tag{1}$$

where $X$ is the conversion efficiency of toluene, and $C_{in}$ and $C_{out}$ are the toluene concentrations in the inlet and outlet gas streams, respectively.

Meantime, the yield of $CO_2$, is resulted from Equation (2):

$$Y = \frac{C_{CO_2}}{6C_{in}} \times 100\% \tag{2}$$

where $Y$ is the yield of $CO_2$, $C_{CO2}$ is the outlet $CO_2$ concentration.

For reaction products, the concentration of $CO_2$ from the outlet stream was measured on the GC-FID equipped with a nickel converting unit. Trace organic byproducts were trapped in a Carbopack B tube (Camsco, Huston, TX, USA) and then detected on a Gas Chromatograph Mass Spectrometer (Agilent 7890B-7000C, St. Clara, CA, USA) coupled with the Thermal Desorption (Markes, Unity-xr, London, UK) instrument. The reaction intermediate was monitored by Fourier transform spectroscopy (DRIFTS, Thermo Nicolet 6700, Waltham, MA, USA), and the experiment and instrument information are shown in Supplementary Material.

### 4. Conclusions

Manganese dioxides with different phases types ($\alpha$, $\beta$, $\delta$, and $\gamma$) were successfully composited with $LaMnO_3$ perovskite by a simple one-pot synthesis route. Combined with the characterization results of SEM, $N_2$ physisorption, XPS and $H_2$-TPR, the upgraded catalytic activity of the most active $\alpha$-$MnO_2$/$LaMnO_3$ should be attributed to its high content of lattice oxygen, better low-temperature reducibility and larger surface area. The activation degree of lattice oxygen on the surface of $MnO_2$/$LaMnO_3$ can be controlled by adjusting the phase-type of $MnO_2$. In addition, the byproducts from the low-temperature oxidation of toluene over $\alpha$-$MnO_2$/$LaMnO_3$ mainly include benzyl alcohol, benzaldehyde, benzoic acid, benzene, acetic acid, etc. This study also provides a method to control the crystal phase of manganese dioxide on the surface of perovskite and a demonstration to reveal the structure-activity relationship.

**Supplementary Materials:** The following supporting information can be downloaded at: https://www.mdpi.com/article/10.3390/catal12121666/s1, Figure S1: XRD patterns of LMO; Figure S2: Toluene oxidation over of (a) $\alpha$-MO and $\alpha$-MO/LMO, (b) $\beta$-MO and $\beta$-MO/LMO, (c) $\delta$-MO and $\delta$-MO/LMO, (d) $\gamma$-MO and $\gamma$-MO/LMO; Text S1: Chemicals and materials; Text S2: Characterization; Text S3: Preparation of $LaMnO_3$; Text S4: Preparation of $MnO_2$ with different crystal phases; Text S5: Normalized reaction rate equation [45–47].

**Author Contributions:** L.L. (Lu Li): Investigation, Data curation, Visualization, Writing—original draft, Funding acquisition; Y.L. and J.L.: Software, Supervision, writing—review & editing; B.Z.: Investigation, Validation, Methodology; M.G. and L.L. (Lizhong Liu): Resources, Funding acquisition, Project administration, Supervision, writing—review & editing. All authors have read and agreed to the published version of the manuscript.

**Funding:** This research was funded by the Natural Science Fundamental Research Project of Jiangsu Colleges and Universities of China (Grant No. 22KJB610022) and Jiangsu Provincial Key Research and Development Program (Grant No. BE2022767).

**Data Availability Statement:** Not applicable.

**Acknowledgments:** The authors of this work appreciate the technical support from the Instrumental Analysis Center of Xi'an Jiaotong University and Nantong University Analysis & Testing center.

**Conflicts of Interest:** The authors declare no conflict of interest.

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
