# Peer review of "Catalytic Degradation of Toluene over MnO2/LaMnO3: Effect of Phase Type of MnO2 on Activity"

_catalysts, doi:10.3390/catal12121666_

Round 1
Reviewer 1 Report
Volatile Organic Compounds emitted from industrial processes and automobile exhaust emissions represent a serious environmental problem. The paper reports interesting findings especially on toluene total oxidation over different phases of MnO2 on LaMnO3.
The manuscript is relevant to the scope of the journal. The paper is well written but some informations are missing and the level of English is good. The paper could be accepted after major revisions because there are some points where the paper must be improved, especially with the following corrections in bold:
*For the catalytic evaluation, the authors write that the air stream contains approximately 1000ppm of toluene. (line 95) What is the precision of the quantities because these will affect the results? And you should give the GHSV. How is the conversion calculated? This does not seem to take into account the total conversion to CO2. For the total oxidation of VOCs, it is important to also have the CO selectivity. What is it?
*Figure 1 presenting the XRD pattern is not very readable. The peaks showing the different alfa, beta, … phases are not intense enough to confirm that these phases have been obtained correctly. You should at least make enlargements on these characteristic peaks.
*For the SEM study, it would be interesting to have the EDX results on the MnO2 particles to confirm the presence of these particles.
*For the catalytic performances, the order of the performances of the catalysts remain unchanged between the T90, T50 and T20 for the catalysts presented in figure 3 therefore the T50 are enough (table 1). Also, the light off curves for the alpha, beta, gamma and delta MO catalysts should be added in figure 3 (or in a separate figure) and would have been nice to have the temperatures of the total conversions for all the MO samples/ LMO.
*The isotherms given in figure 4 do not really allow to conclude on the types of pores and especially the pore size distributions given are not significant to conclude.
*For the XPS study, it would be necessary to give a more complete table with the important reports to make a more developed study; figure 5 is not sufficient.
*Regarding the results in Table 2, the SBET are too precise. Values after the decimal must not be given (insignificant values).
*The study of possible by-products is interesting because the toxicity of these by-products, even emitted in very small quantities, can make the purification process poor. A reference in this field could be added (Co-Mg-Al oxides issued of hydrotalcite precursors for total oxidation of volatile organic compounds. Identification and toxicological impact of the by-products ; C. Gennequin et al; C. R. Chimie, 13, Issue 5, 494 - 2010). For the result in figure 7, it would be necessary to know the conditions for obtaining the result (equipment, temperature or area where the products were taken, etc.)
*The bibliography should not be given exclusively for Chinese authors. Even if there are many good articles by these authors, it is necessary to diversify the sources.Examples:
-Boosting VOCs elimination by coupling different techniques
R. El Khawaja et al.; Chemical Synthesis; 2: 13 (2022) 75 p (31)
-A new approach in the one-step synthesis of a-MnO2 via a modified solution combustion procedure
M. M. Moqaddam et al. Nanoscale Adv., 2022, 4, 3909
-Effect of metal-doping (Me = Fe, Ce, Sn) on phase composition, structural peculiarities, and CO oxidation catalytic activity of cryptomelane-type MnO2; T. S. Kharlamova et al. Journal of Alloys and Compounds Volume 917, 5 October 2022, 165504
-Catalytic combustion of VOCs over a series of manganese oxide catalysts
S. C. Kim et al Applied Catalysis B-environmental 2010
-Catalytic Abatement of Volatile Organic Compounds and Soot over Manganese Oxide Catalysts; Miguel Jose Marin Figueredo et al; Materials 2021
-Acid washing of MnOx ‐ SBA‐15 composites as an efficient way to improved catalytic properties in HCHO total oxidation, G. Rochard et al; ChemNanoMat 6(8) 2020
*There are a few typographical errors; for example:
line 133: there is twice "should"
line 254: there is an extra e before oxidation: "eoxidation".
line 273: a space should be removed.
Author Response
Dear Editor and Reviewers:
Thank you for your letter and the reviewers’ comments concerning our manuscript entitled ”Catalytic oxidation of toluene over MnO2/LaMnO3: Effect of phase type of MnO2 on activity” (Manuscript ID: catalysts-2062397). We sincerely thank the editor and all reviewers for their valuable feedback that we have used to improve the quality of our manuscript. We have studied the comments very carefully and tried our best to revise our manuscript. The reviewer comments are laid out below and specific concerns have been numbered. Our response is given in blue text and the changes/additions to the manuscript are given in yellow highlight. According to the reviewers’ nice suggestions, we have made extensive corrections to our previous draft, the point-by-point responses and detailed corrections are listed below.
Reviewer #1:
Volatile Organic Compounds emitted from industrial processes and automobile exhaust emissions represent a serious environmental problem. The paper reports interesting findings especially on toluene total oxidation over different phases of MnO2 on LaMnO3.
The manuscript is relevant to the scope of the journal. The paper is well written but some informations are missing and the level of English is good. The paper could be accepted after major revisions because there are some points where the paper must be improved, especially with the following corrections in bold.
Response: Thank you very much for the positive evaluation and professional comments.
- For the catalytic evaluation, the authors write that the air stream contains approximately 1000ppm of toluene. (line 95). What is the precision of the quantities because these will affect the results? And you should give the GHSV. How is the conversion calculated? This does not seem to take into account the total conversion to CO2. For the total oxidation of VOCs, it is important to also have the CO selectivity. What is it?
Response: Thank you for the professional suggestion. The concentration of toluene was strictly controlled by a mass flowmeter, where all the inlet gas streams of the whole reaction system are controlled by different sensing ranges of mass flowmeters (Sevenstar co., China), with the detection accuracy of ±1.0% S.P. (≥ 35% F.S), and ±0.35% S.P. (< 35% F.S). Additionally, the original toluene was purged from standard gas in the gas cylinder, with 1% ± 0.03% of standard conditions. All mentioned above could strictly control the precision of the concentration of toluene, the concentration error is ±3 ppm.
Due to the comparison with other similar catalysts in Table S1 of Supplementary Material, we use the index of the weight hourly space velocity (WHSV) value, which was also added in section “2.2. Catalytic evaluation” as “with a weight hourly space velocity (WHSV) of ca. 12 000 mL/(g h-1)” and highlights in yellow color.
The toluene conversion efficiency equation has been replenished in section “2.2. Catalytic evaluation” and marked in yellow.
This study mainly focused on the effect of phase type of MnO2 in LaMnO3.15 on the total catalytic degradation of toluene, and we have tested the CO2 yields in our experiments and added in the article as Fig. 3b and Table 1, and the corresponding description was also added in section “3.2. Catalytic performance” as “To better evaluate the destruction ability of prepared catalysts, the mineralization efficiency of toluene oxidation was also measured. As shown in Fig 3b and Table 1, the mineralization temperatures of these catalysts for toluene are higher than that of conversion, where the variation trend of CO2 yield was similar to that of conversion. Besides, α-MO/LMO sample still maintains the best behavior in CO2 generation capacity.”, which is highlighted in yellow. Meantime, the equation to calculate CO2 yield efficiency was also replenished as eq. 2 in section “2.3. Catalytic evaluation” and marked in yellow color. As for the research on CO selectivity, we will make further exploration in the future. In addition, the title of the manuscript was revised accordingly as "Total catalytic degradation of toluene over MnO2/LaMnO3: Effect of phase type of MnO2 on activity".
- Figure 1 presenting the XRD pattern is not very readable. The peaks showing the different alfa, beta, … phases are not intense enough to confirm that these phases have been obtained correctly. You should at least make enlargements on these characteristic peaks.
Response: Thank you so much for the comment. As shown in the Fig. S1 of the Supplementary Information document, these characteristic peaks have been magnified between 2Ɵ of 10–40°, which is supplemented in “3.1. Morphology and crystal phase structure” as “with the magnified view in Fig. S1 of Supplementary Information, it was found that” and marked in yellow.
- For the SEM study, it would be interesting to have the EDX results on the MnO2 particles to confirm the presence of these particles.
Response: Thank you for the suggestion. Although the Energy Dispersive X-Ray Spectroscopy (EDX) in SEM can test the surficial element type, it can only reflect the distribution of elements, in another word, it cannot confirm the crystal or compound of the represented element. Not only MnO2 particles will be identified, but also the Mn cation from LaMnO3.15 surface, so it still cannot identify which part belongs to supported MnO2 particles, and which part reflect Mn from LaMnO3.15. The best way to recognize the MnO2 crystal and its phase still depended on XRD patterns. To present the phase detail of different MnO2 phases more clearly, we have enlarged the XRD pattern range in Fig. S1 (Supplementary Information) and confirmed the existence of MnO2 with different crystal structures.
- For the catalytic performances, the order of the performances of the catalysts remain unchanged between the T90, T50 and T20 for the catalysts presented in figure 3 therefore the T50 are enough (table 1). Also, the light off curves for the alpha, beta, gamma and delta MO catalysts should be added in figure 3 (or in a separate figure) and would have been nice to have the temperatures of the total conversions for all the MO samples/LMO.
Response: Thank you for the professional suggestion. The T90 and T20 index were removed and the light-off curves for the α, β, γ, and δ MnO2 catalysts have been provided in the Supplementary Material as Fig. S2.
- The isotherms given in figure 4 do not really allow to conclude on the types of pores and especially the pore size distributions given are not significant to conclude.
Response: Thank you for your kind reminder and comment. The description of the types of pores and the pore size distributions and Fig. 4b have been deleted. The isotherm curves have been merged with XRD patterns in Fig. 1, which was presented as “Fig. 1 (a) XRD patterns and (b) N2 adsorption-desorption isotherms of LMO, α-MO/LMO, β-MO/LMO, δ-MO/LMO, and γ-MO/LMO samples.”, the figure number in the manuscript was also revised, all changes were marked in yellow.
- For the XPS study, it would be necessary to give a more complete table with the important reports to make a more developed study; figure 5 is not sufficient.
Response: Thank you. In fact, we have been provided with the more complete Table with the important reports in Table 2 of the original Manuscript.
- Regarding the results in Table 2, the SBET is too precise. Values after the decimal must not be given (insignificant values).
Response: Thanks for your kind reminder. The significant digits of SBET have been reasonably readjusted into 3 digits.
- The study of possible by-products is interesting because the toxicity of these by-products, even emitted in very small quantities, can make the purification process poor. A reference in this field could be added (Co-Mg-Al oxides issued of hydrotalcite precursors for total oxidation of volatile organic compounds. Identification and toxicological impact of the by-products; C. Gennequin et al; C. R. Chimie, 13, Issue 5, 494 - 2010). For the result in figure 7, it would be necessary to know the conditions for obtaining the result (equipment, temperature or area where the products were taken, etc.)
Response: Thanks for your professional suggestion. The reference focused on toxicity in humans provides a wide view of the VOC oxidation elimination field, which could more precisely evaluate the technique from the point of view of health risk assessment. We think the comment makes inspiration for the application safety of candidature catalysts or technique study, which would be our future research direction. In this manuscript, we added the point of future vision at the end of the GC/MS part as “Even though a higher degradation efficiency was found for α-MO/LMO catalyst and only trace organic byproducts were detectable in the outgas, some of the products would cause more toxicity to human health or environment [41], even in a very small quantity. The comprehensive assessment of these trace byproducts would be our future research direction to help the choice of new-dedicated catalysts for the VOC oxidation technique.”, which was highlighted in yellow color.
The organic byproducts measurement information has been added in section “2.3. Catalytic evaluation” as “Trace organic byproducts were trapped in a Carbopack B tube (Camsco, USA) and then detected on a Gas Chromatograph Mass Spectrometer (Agilent 7890B-7000C, USA) coupled with the Thermal Desorption (Markes, Unity-xr, UK) instrument.” and highlighted in yellow color. The trace organic compounds were sampling under the T50 reaction condition of α-MO/LMO oxidation experiment, which was added in “3.6 Possible degradation mechanism” as “trace byproducts from the T50 condition of α-MO/LMO oxidizing toluene illustrated” and marked in yellow color.
- The bibliography should not be given exclusively for Chinese authors. Even if there are many good articles by these authors, it is necessary to diversify the sources. Examples:
-Boosting VOCs elimination by coupling different techniques. R. El Khawaja et al.; Chemical Synthesis; 2: 13 (2022) 75 p (31)
-A new approach in the one-step synthesis of a-MnO2 via a modified solution combustion procedure. M. M. Moqaddam et al. Nanoscale Adv., 2022, 4, 3909
-Effect of metal-doping (Me = Fe, Ce, Sn) on phase composition, structural peculiarities, and CO oxidation catalytic activity of cryptomelane-type MnO2; T. S. Kharlamova et al. Journal of Alloys and Compounds Volume 917, 5 October 2022, 165504
-Catalytic combustion of VOCs over a series of manganese oxide catalysts. S. C. Kim et al Applied Catalysis B-environmental 2010
-Catalytic Abatement of Volatile Organic Compounds and Soot over Manganese Oxide Catalysts; Miguel Jose Marin Figueredo et al; Materials 2021
-Acid washing of MnOx ‐ SBA‐15 composites as an efficient way to improved catalytic properties in HCHO total oxidation, G. Rochard et al; ChemNanoMat 6(8), 2020
Response: Thank you for the suggestion. Some of the references were accepted and added in a certain area of the manuscript as [19] and [20], which were highlighted in yellow color.
- There are a few typographical errors; for example:
line 133: there is twice "should"
line 254: there is an extra e before oxidation: "eoxidation".
line 273: a space should be removed.
Response: We were really sorry for our careless mistakes. All the errors were revised.
Reviewer 2 Report
English should be noticeably improved.
Introduction should be revised.
X-ray data as presented are not clear. Peaks of MnO2 phases in the Figure 1 are negligible, on background level. Evidence of a presence of different modifications of MnO2 phases in the samples should be demonstrated more clearly.
What is the catalysts particle size in catalytic experiments?
What is the MnO2 content in the samples? Is MnO2 content the same in different samples?
What is the normalized (per m2) activity of the samples? What is the reason for different activity of different MnO2 modifications?
Description of H2-TPR data should be revised taking into account MnO2 content in the samples.
Round 2
Reviewer 1 Report
The corrections have been taken into consideration. There is just figure 1S (XRD) which still does not present well the 2theta characteristic of the alpha, beta phases...
Reviewer 2 Report
Dimensions in eq. S1 should be indicated.
